# Enriching Facial Anti-Spoofing Datasets via an Effective Face Swapping Framework

**DOI:** 10.3390/s22134697

**Published:** 2022-06-22

**Authors:** Jiachen Yang, Guipeng Lan, Shuai Xiao, Yang Li, Jiabao Wen, Yong Zhu

**Affiliations:** School of Electrical and Information Engineering, Tianjin University, Tianjin 300072, China; yangjiachen@tju.edu.cn (J.Y.); lgp@tju.edu.cn (G.L.); liyang328@shzu.edu.cn (Y.L.); wen_jiabao@tju.edu.cn (J.W.); yongzhu@tju.edu.cn (Y.Z.)

**Keywords:** multimedia security, facial anti-spoofing, biomedical big data, generative adversarial network, latent feature analysis

## Abstract

In the era of rapid development of the Internet of things, deep learning, and communication technologies, social media has become an indispensable element. However, while enjoying the convenience brought by technological innovation, people are also facing the negative impact brought by them. Taking the users’ portraits of multimedia systems as examples, with the maturity of deep facial forgery technologies, personal portraits are facing malicious tampering and forgery, which pose a potential threat to personal privacy security and social impact. At present, the deep forgery detection methods are learning-based methods, which depend on the data to a certain extent. Enriching facial anti-spoofing datasets is an effective method to solve the above problem. Therefore, we propose an effective face swapping framework based on StyleGAN. We utilize the feature pyramid network to extract facial features and map them to the latent space of StyleGAN. In order to realize the transformation of identity, we explore the representation of identity information and propose an adaptive identity editing module. We design a simple and effective post-processing process to improve the authenticity of the images. Experiments show that our proposed method can effectively complete face swapping and provide high-quality data for deep forgery detection to ensure the security of multimedia systems.

## 1. Introduction

With the development of Internet of things technologies [1,2], deep learning [3,4,5], and communication technologies [6,7], wearable devices and intelligent terminal devices have developed rapidly, which have brought unprecedented changes to many fields. Intelligent medical systems, intelligent industrial systems, and smart home systems are changing people’s lifestyles. At the same time, they also speed up data collection, processing, and transmission [8]. However, in the era of big data, it is inevitable that it will cause malicious tampering and forgery of data, which poses a potential threat to personal privacy and public opinion [9]. Especially when the deep learning technologies gradually mature, various forgery technologies have achieved realistic results. Taking personal portraits as an example, face forgery technologies cannot be detected by the human eye, as shown in Figure 1. Many countries and regions actively have carried out the improvement of relevant regulations and laws. Portrait data, especially in the digital medical system, is extremely sensitive personal data. Improving privacy and security has become one of the hot topics.

With the development of deep learning technologies [10,11,12], great progress has been made in image understanding [13,14], image generation [15,16] and image classification [17]. With the development of deep face forgery technologies [18,19], researchers also carry out deep forgery detection technologies [20,21] at the same time. However, most of the current deep forgery detection technologies are learning-based methods. These methods rely heavily on data. Diversification and sufficient data is one of the necessary means to improve the generalization and accuracy of deep forgery detection methods. Due to the sensitivity of human eyes to familiar structures, generating high-quality face images that meet the downstream tasks is very challenging work.

In order to improve portrait security in social media, we propose an effective face swapping framework to enrich facial anti-spoofing datasets. Firstly, we utilize the feature pyramid network to extract the latent codes of the images. Secondly, the identity weight generation module is used to increase the attention to the identity information in the latent codes. Thirdly, we propose an adaptive identity editing module, which is based on the AdaIN [22] mechanism and can realize identity transformation. Fourth, through the generator of StyleGAN [15], the edited latent codes can be decoded into high-quality face images. Finally, in order to improve the overall authenticity of the images, we propose a post-processing method. In this work, our main contributions are as follows:We propose an effective face swapping framework, which can generate photorealistic results.We use the identity weight generation module to increase the attention to the identity information.We propose an adaptive identity editing module to realize the identity transformation.We utilize post-processing to improve the authenticity, and experiments verify the efficiency of our face swapping framework.

The rest of this paper is arranged as follows: we introduce the related works of this paper in Section 2. In Section 3, we introduce the method in detail, including the feature pyramid network, mapping network, identity weight generation module, adaptive identity editing module, generator, post-processing process, and loss functions. In Section 4, we show the results of the proposed face swapping framework, and design ablation experiments and comparative experiments to verify the efficiency of our proposed method. Finally, in Section 5, we elaborate on our conclusions and the focus of future work.

## 2. Related Works

In this section, we introduce the related works of this paper from four aspects. In Section 2.1, we introduce the fundamental of the paper, that is, the generative adversarial network. In Section 2.2, we introduce the deep face forgery detection methods and the composition of existing face forgery datasets. In Section 2.3, we briefly introduce the latent code operation based on the latent space. In Section 2.4, we report the related works of face forgery methods.

### 2.1. Generative Adversarial Network

Goodflow et al. [3] proposed the generative adversarial networks in 2014. The generative adversarial network consists of a generator and a discriminator. The function of the generator is to generate sufficiently real images, and the function of the discriminator is to identify whether the input image is from the real image (i.e., the images in the dataset) or from the fake images (i.e., the images generated by the generator). The generator and the discriminator are against each other during the training process, and finally, the discriminator cannot distinguish the authenticity of the images. The special structure and training method of the generative adversarial network ensure the authenticity of the generation, and then it is widely used in image generation [15], image editing [18,23], and other fields [24,25].

### 2.2. Deep Face Forgery Detection Methods and Datasets

Since deepfake was developed in 2017, face forgery videos or images based on deep learning have emerged on the Internet. Relevant statistics show that more than 96% of face forgery videos and images are illegal, which poses a potential threat to personal privacy and public opinion. At the same time, deep forgery detection technologies are also developing. Tariq et al. [26] proposed a forgery images forensics platform, FakeFaceDetect. The platform uses dual stream faster R-CNN to capture the high-level information and low-level information of the images, so as to realize the detection of various forgery facial images. Yang et al. [27] found that there are subtle texture differences in image saliency between real images and forgery images for the first time. They used the guided filter with saliency mapping as the guidance mapping, showing the potential characteristics of forgery facial images. Yu et al. [28] found that different generative adversarial networks have specific fingerprints, and these specific fingerprints are left in the generated images. Yu et al. utilized this feature to distinguish between real images and forgery images. Yang et al. [21] found that multi-scale texture information played an important role in depth forgery detection in the process of exploring CNN to distinguish true and fake images. Based on this discovery, they proposed MTD-Net for deep forgery detection by using central difference convolution and atrous spatial pyramid pooling. Qian et al. [29] introduced the frequency domain into forgery detection. They found that the frequency domain provided a complementary viewpoint of forgery artifacts and can better describe compression errors, and then they proposed a novel frequency in the face forgery network.

At present, deep forgery detection methods are mostly learning-based methods, which rely heavily on data. Commonly used face forgery datasets include Faceforensics++, Celeb-DF [30], Deeperforensics [31] and DFDC [32]. Faceforensics++ includes fake face videos and real face videos. The fake videos were generated by four face forgery methods. The methods of forgery face videos include Deepfakes, Faceswap, Face2Face, and NeuralTexture. Faceforensics++ divides the quality of forgery videos from low to high into three levels: raw, c23, and c40. Celeb-DF contains 5639 high-quality deepfake videos and 590 real videos, corresponding to more than 2 million frames. Real videos were collected from YouTube, including 59 characters, which are diverse in gender, age, posture, and so on. Deepforensics collected 100 face images and 1000 public videos on YouTube. They swapped each image to the corresponding 10 videos, and 1000 forgery videos were obtained. DFDC is a large-scale face forgery dataset published by Facebook and Microsoft, which contains 100,000 forgery video clips. These forgery videos are obtained by deepfake, GAN-based methods, and non-learned methods.

### 2.3. Latent Code Editing

Since the proposal of convolutional neural networks, researchers have found that the shallow features extracted by convolutional networks have low-level semantic information, such as structure, color, and so on, while the deep features have high-level semantic information, such as identity information and so on. In addition, researchers found that editing images through latent codes could achieve better results. After Ian et al. [3] proposed that after generative adversarial network, the latent space editing based on GAN has achieved a great leap. Härkönen et al. [33] found the latent direction of attributes in the latent space based on principal component analysis. Shen et al. [34] first found that generative adversarial networks learned semantics in linear subspaces of the latent space. After the proposal of StyleGAN, due to the powerful generation ability of StyleGAN and the feature de-entanglement ability in the latent space of StyleGAN, a large amount of latent space editing work is set to be completed in the latent space of StyleGAN. Emily et al. [35] utilized the fully supervised method to realize latent code editing in StyleGAN for the first time to realize the change of attributes. Or et al. [36] used the prior knowledge in natural language processing to find the relationship between features in latent space and text information.

### 2.4. Face Forgery Technologies

According to the effect of face forgery generated by different methods, face forgery technologies can be divided into three categories: face reenactment [37,38], face swapping [17,18], and face local editing [39]. Facial reenactment refers to changing only the expression and posture attributes in the whole facial image while keeping other attributes unchanged. The specific effect is shown in the first line of Figure 1. Face swapping refers to changing only the identity information in the whole facial image while keeping other attributes unchanged. The effect is shown in the second line of Figure 1. Face local editing refers to changing only a partial region of the facial image while maintaining other regions unchanged, such as changing the hairstyle, changing the hair color, changing eye shape, etc.

The early face swapping works carry out face cutting and face fusion on the basis of similar posture. These kinds of methods have great limitations and cannot complete the face swapping in the case of large skin color differences and large posture differences. With the proposal of convolutional neural networks and generative adversarial networks, the above problems are gradually solved through the extraction and fusion of high-level features and low-level features. Güera et al. [40] used CNN to extract frame-level information and used RNN to judge whether a subject needed to be operated in each frame. Natsume et al. [41] proposed RSGAN, which independently processed the facial region and hair region in the latent space, and realized face swapping by replacing the latent spatial representation of the facial region. Nirkin et al. [42] introduced a continuous interpolation of the face viewpoints based on reenactment, Delaunay triangulation, and barycentric coordinates and realized face swapping with subject agnostic. Li et al. [19] proposed a dual-stream network, which provided attribute information and identity information respectively. Through the proposed fusion model, high-fidelity results are achieved. Zhu et al. [43] realized one-shot face swapping by using the face transfer module with the help of the powerful generation ability of StyleGAN.

## 3. Method

In this section, we introduce the method proposed in this paper in detail. The section is divided into five parts. First, we explore the position of identity information in latent codes, which will affect subsequent decisions on how to edit latent codes. In Section 3.2, we briefly introduce the encoder, mapping network, and generator. In Section 3.3, we focus on the identity editing module, which is the key to achieving accurate identity transformation. In Section 3.4, we introduce the post-processing process proposed in this paper, which will improve the authenticity of the generated images. In Section 3.5, we report the loss functions.

The overall structure of the network is shown in Figure 2. The network is composed of a feature pyramid network with ResNet, mapping network, identity weight generation module, adaptive identity editing module, generator, and post-processing process. For the convenience of representation, we unify the representation of symbols here. We use $I_{id}$ to represent the image that provides identity information, and $I_{att}$ to represent the image that provides attributes other than identity information. The output of the StyleGAN generator is represented by $I_{result}$. After the post-processing process, the final generation result is represented by $I_{final}$.

### 3.1. Exploration and Motivation

It is well known that the latent codes of an image contain all the features of the corresponding image. These features can be roughly divided into two categories: fine-grained features, such as identity information, expression information, and posture information, and coarse-grained features, such as color information and structure information. In order to design the subsequent adaptive identity editing module, we explore the location of identity information in latent codes. As shown in Figure 2, we use an encoder, mapping network, identity weight generation module, and generator consistent with the structure of the face swapping process.

In the latent space, the specific location of these features is uncertain, and the features are entangled with each other, which causes a great degree of confusion. In order to edit only the features that we want to edit and prevent the interference of other features as much as possible, we need to extract the area to be edited. At this stage, researchers have done too little research on the structural level of latent space, so they can not accurately determine the specific location of the corresponding features. Therefore, we propose an identity weight generation module, which can adaptively find the location of the area to be edited through a large number of data training and weight updating.

Different from the face swapping process, the function of the weight generation module here is to obtain a weight map between 0 and 1, which is used to visualize the position of identity information in latent codes. The loss functions in this process are consistent with the loss functions in the face swapping process, but the accuracy of the identity information transformation is not required (we introduce the loss function in detail in Section 3.5). In the identity transformation stage of the exploration and motivation process, the obtained mask is multiplied by the latent codes mapped into StyleGAN to obtain the region of interest, and simple feature fusion is realized by element addition. Finally, the fused latent codes are input into the generator to get the result of changing only the identity information.

When the network training converges, we visualize the output value of the identity weight generation module to obtain the heat map shown in Figure 3.

From Figure 3 we can see that the distribution of identity information in latent codes does not have a certain regularity, and the content of identity information in each latent vector is also different. We believe that the reason for this phenomenon is that the image is in the wild, which is different from the image generated by a neural network. Each feature is entangled with others, resulting in the global distribution of identity information in latent codes.

### 3.2. Encoder, Mapping Network, and Generator

In order to extract the latent features of the image more comprehensively, we use the feature pyramid structure [44] with ResNet [45] as the encoder. In order to meet the requirements of StyleGAN, according to the resolution of the input image, we choose (i.e., 256 × 256), we set the dimension of encoder output is 14 × 512. The latent space of the latent code output by the encoder and the w+ latent space of StyleGAN are two different latent spaces. In order to map the latent code obtained by the encoder only to the latent space of StyleGAN, we set up the mapping network. The main structure of the mapping network is the linear layer, and each latent vector corresponds to a mapping sub-network, that is, the mapping network is composed of 14 mapping sub-networks. We use the StyleGAN generator as the generator of our framework. With the help of the powerful generation ability of StyleGAN and the face generation prior knowledge, we can get high-quality results.

### 3.3. Identity Editing Module

We divide the latent codes editing stage into two parts: identity weight generation and adaptive identity editing.

#### 3.3.1. Identity Weight Generation

The weight generation module is composed of multiple 1×1 convolution and the sigmoid activation function, whose function is to add an identity weight on the basis of the original latent code. For the latent code of $I_{id}$, it pays more attention to the region representing identity information, and for the latent code of $I_{att}$, it pays more attention to the feature other than identity information. In the network, there are two identity weight generation modules. They have the same structure. They are responsible for extracting the regions of interest of identity information and attribute information respectively. The process is as follows:(1)$$mask_{id}=\mathrm{IWG}\left(L_{id}\right)$$
(2)$$mask_{att}=\mathrm{IWG}\left(L_{att}\right)$$
(3)$$RoI_{id}^{id}=mask_{id}\times L_{id}$$
(4)$$RoI_{att}^{att}=mask_{att}\times L_{att}$$
where IWG represents the identity weight generation module, $L_{id}$ and $L_{att}$ represent the latent codes of $I_{id}$ and $I_{att}$ respectively. $RoI_{id}^{id}$ represents the latent code after enhancing the identity information in the latent code of $I_{id}$.

#### 3.3.2. Adaptive Identity Editing

The adaptive identity editing module is the key to realizing identity transformation in this paper. The module uses the identity information extracted from the existing face recognition network to change the specific information in the latent codes. We use the identity information extracted from the face recognition network to fix the direction of the latent codes. The adaptive mask in this module further extracts the feature regions that need to be edited and reduces the overall effect decline caused by changes to other latent features.

The structure of the adaptive identity editing module is shown in Figure 4. The structure is based on the AdaIN [22] mechanism and the adaptive blending mechanism.

The main structure of the adaptive identity editing module is divided into three parallel channels, and the output result is the increment of identity transformation. The process is as follows:

Step 1: Input $RoI_{id}^{id}$ into the intermediate channel, that is, the adaptive mask generation path. The adaptive mask generation module is composed of two 1×1 convolutions and a sigmoid activation function. The process is as follows:(5)$$mask=\mathrm{AMG}(RoI_{id}^{id})$$
where AMG is the adaptive mask generation module.

Step 2: Input $RoI_{id}^{id}$ into the upper path, which is to edit the identity information in the latent code. The identity editing module is composed of four identity editing blocks based on AdaIN. Because there are no learnable parameters in the AdaIN mechanism, we design a residual structure for learning identity information editing. The process is as follows:(6)$$L_{edit}^{id}=\mathrm{IEB}\left(RoI_{id}^{id}\right)$$
(7)$$L_{out}=\left(\frac{L_{in}-mean\left(L_{in}\right)}{std(L_{in})+eps}\times std\left(v_{id}\right)\right)+mean\left(v_{id}\right)$$
where IEB is the identity editing blocks. $L_{in}$ and $L_{out}$ are the input and output of the AdaiN layer respectively. $v_{id}$ is the output of the face recognition network. mean is the process of calculating the average value and std is the process of calculating the variance. eps = 10 × 10${}^{-5}$ to prevent the denominator from being 0.

Step 3: Use the adaptive mask obtained in Step 1 to eliminate irrelevant features, and finally only retain the increment of identity transformation. The process is as follows:(8)$$\Delta id=mask⊗L_{edit}^{id}⊕\left(1-mask\right)⊗RoI_{id}^{id}$$

The reason why we set the adaptive mask in the adaptive identity editing module is to retain other information related to identity information in $L_{edit}^{id}$, such as location information. At the same time, the information irrelevant to identity information, such as color information, is eliminated in $RoI_{id}^{id}$. Finally, the edited latent codes are shown below:(9)$$L_{edit}^{all}=RoI_{att}^{att}+\Delta id$$
where $L_{edit}^{all}$ is the edited latent code, then $L_{edit}^{all}$ is input into the StyleGAN’s generator, and we can get high-quality results.

### 3.4. Post-Processing

In order to achieve high-resolution face generation, we use StyleGAN’s generator as the face generator of the method proposed in this paper. However, there is a defect in using the prior knowledge of the face, that is, the generated images only focus on the face region, and often ignore the hair region and the background region. This greatly reduces the authenticity of the generated image. To solve the above problems, we use the existing face parsing to get the face semantic segmentation mask, use the mask to weight the generated image and the image that provides other attributes, and finally get the photo-realistic results. The specific steps are as follows:

Step 1: Firstly, we use the face parsing network to get $I_{att}$’s face parsing map. The binary mask is obtained by classifying the foreground and background of the face parsing map.

Step 2: Because the result of the direct combination has sharp edges, we use Gaussian blur to process the binary mask to obtain $mask_{gua}$ to solve this problem. In this way, a transition area will be formed at the edge of the foreground and background, which will eliminate the above sharp edges.

Step 3: We use $mask_{gua}$ to combine $I_{att}$ and $I_{result}$ to get the final result $I_{final}$.
(10)$$I_{final}=I_{att}⊗\left(1-mask_{gua}\right)⊕I_{result}⊗mask_{gua}$$

### 3.5. Loss Functions

In this work, we set identity loss $L_{id}$, self-reconstruction loss $L_{recon}$, and attribute loss $L_{att}$ as the loss functions of this paper. In the exploration stage and the face swapping stage, we use the same loss functions to train the network.

#### 3.5.1. Identity Loss

In order to ensure that the identity information of $I_{result}$ is consistent with that of $I_{id}$, we set identity loss. We use face recognition network as the extractor of identity information (in this paper, we use ArcFace [46] as the extractor). Identity loss is shown as follows:(11)$$L_{id}=cos\left(\mathrm{extractor}\left(I_{id}\right),\mathrm{extractor}\left(I_{result}\right)\right)$$
where cos represents cosine similarity loss, and extractor represents face recognition network.

#### 3.5.2. Self-Reconstruction Loss

Because there is no ground truth in the face swapping task, there is no strong constraint in the training process. In order to enhance the robustness and stability of the network, we propose the self-reconstruction loss. We set a self-reconstruction every four times. When the network is self-reconstruction, $I_{id}$ = $I_{att}$. The self-reconstruction loss is shown as follows:(12)$$L_{recon}=L_{1}\left(I_{id},\mathrm{net}\left(I_{id},I_{id}\right)\right)$$
where $L_{1}$ is the l1 loss. Through the above process, we can achieve strong constraints in the training network, which greatly improves the stability and robustness of network training.

#### 3.5.3. Attribute Loss

In order to ensure that the attributes are consistent with $I_{att}$ during face swapping, we set attribute loss. We use VGG-19 [47] loaded with pre-training parameters as an attribute feature extractor. Some of the feature maps are taken for constraints to ensure the consistency of attribute information.
(13)$$L_{att}=\frac{1}{L}\sum_{i=1}^{L}\left|\right|F_{i}\left(I_{result}\right)-F_{i}\left(I_{att}\right){\left|\right|}_{2}$$
where L is the number of the feature maps, $F_{i}$ is the *i*-th feature maps extracted by VGG-19.

#### 3.5.4. Objective Function

The objective function of the network is as follows:(14)$$L_{net}=\lambda_{id}\times L_{id}+\lambda_{recon}\times L_{recon}+\lambda_{att}\times L_{att}$$
where $\lambda_{id}$ = 3, $\lambda_{recon}$ = 5 and $\lambda_{att}$ = 3, which are the hyper-parameters respectively.

## 4. Results and Experiments

We introduce this section in five parts: in Section 4.1, we introduce the dataset and experimental setting. In Section 4.2, we briefly introduce the evaluation metrics used for quantitative comparison. In Section 4.3,we display some generation effects of our proposed model. In Section 4.4, we prove the superiority of the adaptive identity editing module. In Section 4.5, we compare our framework qualitatively and quantitatively with other models.

### 4.1. Dataset and Experimental Setting

#### 4.1.1. Dataset

In this work, the dataset we used is CelebA-HQ [48]. The dataset is a large-scale HD face dataset with a resolution of 1024×1024. The dataset is widely used in face recognition, face segmentation, and other face tasks. CelebA-HQ is further developed on the basis of CelebA [49]. CelebA-HQ has 30,000 centered face images through cutting and rotating.

#### 4.1.2. Experimental Setting

The hardware conditions of the experiments are Intel (R) Xeon (R) CPU E5-2620 V4 and two NVIDIA GTX Titan XP GPUs. The whole experiment is implemented on the PyTorch platform. We use Adam [50] as the optimizer of the training process. The epoch of training is 250,000 and the batch size is 1. Model weights and results are saved through hyper-parameters.

### 4.2. Evaluation Metrics

In the quantitative experiment, we take identity similarity, expression similarity, and FID [51] value as quantitative metrics.

#### 4.2.1. Identity Similarity

Similar to identity loss, the formula of identity similarity is as Equation (Equation 15). The larger the identity similarity, the closer the generated image is to the identity information of $I_{id}$
(15)$$\mathrm{id}\;\mathrm{similarity}=cos(\mathrm{extractor}\left(result\right),\mathrm{extractor}\left(I_{id}\right))$$

#### 4.2.2. Expression Similarity

We use a 3D face attribute extraction network to extract expression information. The expression similarity formula is as Equation (Equation 16). The smaller the expression similarity, the smaller the expression similarity between the result and $I_{att}$.
(16)$$\exp\;\mathrm{similarity}=L_{1}\left(result,I_{att}\right)$$

#### 4.2.3. FID

FID is a metric used to measure the quality and diversity of generated images. The smaller the FID value, the better the quality of the generated images.

### 4.3. The Generation Results of Our Model

As shown in Figure 5, we display partial results of our framework. Our proposed framework can realize the maintenance of expression and the transformation of identity and can achieve photo-level authenticity.

### 4.4. Superiority of the Adaptive Identity Editing Module

In order to verify the superiority of the adaptive identity editing module in the process of identity transformation, we set up an ablation experiment. Firstly, we set up the comparison model net-w/o AIE: the network has the same encoder, mapping network, identity weight generation module, and decoder as the face swapping network in this paper but does not include an adaptive identity mapping module. The results generated by the two networks are shown in Figure 6.

#### 4.4.1. Qualitative Analysis

From the qualitative analysis of the figure, we can see that although the generation results of net-w/o AIE can reach photo-realistic authenticity. However, from the perspective of effect and details, there are still many defects. Firstly, in terms of effect, the images generated by net-w/o AIE are not as good as net in identity transformation. We can infer that the identity editing based on the AdaIN mechanism in the adaptive identity editing module is effective. Secondly, in terms of details, the hair color, skin color, and general tone of the background of net-w/o AIE are not consistent with $I_{att}$. Therefore, we can infer that the adaptive mask mechanism plays a key role. In summary, we can infer that the adaptive identity editing module has advantages in identity transformation and attribute maintenance.

#### 4.4.2. Quantitative Analysis

In order to more clearly reflect the superiority of the adaptive identity editing module, we set up a quantitative analysis. We select 30 pairs of images as the objects of quantitative analysis. Calculate the identity similarity and expression similarity of each pair of images respectively and calculate FID between two groups of images. Then the calculated identity similarity and expression similarity are averaged. The results are shown in Table 1.

It can be seen from the Table 1 analysis that when the adaptive identity editing module is removed, the performance of the generated images in identity similarity and expression similarity decreases. Although net-w/o AIE is slightly better than net in FID performance, there is little similarity between the two network generation results.

### 4.5. Comparison with Other Models

We choose FaceSwap, FSGAN [42], SimSwap [52], and Faceshifter [19] as the comparison methods of this paper. The results generated by similarity models are shown in Figure 7.

#### 4.5.1. Qualitative Analysis

It can be seen from Figure 7 that the comprehensive effect of our proposed method on identity transformation and attribute maintenance is the best. In terms of skin color maintenance, some results generated by Faceshifter are better than our proposed method, but they are inferior to our proposed method in identity transformation. We find that there are abnormal facial distortions or irregular color patches in the implementation results of FaceSwap. This is because FaceSwap is completed based on dlib and opencv and lacks the fitting between samples with large differences in posture, skin color, and so on. Although FSGAN has achieved good results in identity transformation, the overall quality is low. After amplification, it will be found that the overall effect is poor compared with other methods. To sum up, our proposed method solves some defects of the existing face swapping methods and significantly improves identity transformation and attribute maintenance.

#### 4.5.2. Quantitative Analysis

In order to clarify the detailed differences between various face swapping methods, we set up a quantitative comparison. We select 30 groups of results generated by different models as the objects of quantitative comparison. Calculate the identity similarity and expression similarity of each pair of images respectively and calculate FID between two groups of images. Then the calculated identity similarity and expression similarity are averaged. The results are shown in Table 2.

From the Table 2 analysis, we can see that our proposed method is the best in the two metrics of identity similarity and FID. Although our effect is slightly inferior to FSGAN in expression similarity, the gap with Faceshifter is relatively small. On the whole, our proposed method has certain efficiency, which can solve some problems of existing methods and provide more realistic and high-definition face forgery images.

## 5. Conclusions and Expectations

In this work, we propose a face swapping framework for enriching facial anti-spoofing datasets. Through experimental verification, we prove the efficiency of our overall framework and the superiority of the adaptive identity editing module. The generated results achieve photo-level authenticity and can be used in social media to improve the accuracy of deep forgery detection and ensure portrait security. In future work, we focus on higher quality and more detailed face swapping results and the privacy and security of multimedia systems.

## Figures and Tables

**Figure 1 sensors-22-04697-f001:**
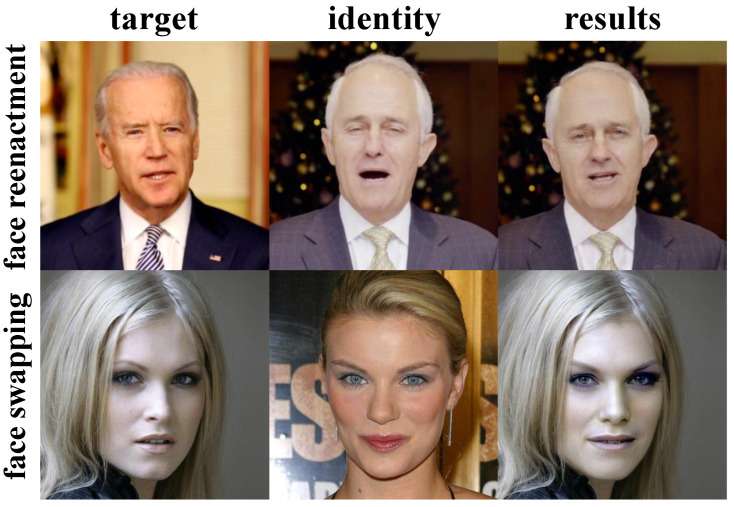
Some technologies realize the results of face forgery. The first line is face reenactment and the second line is face swapping. The result of face swapping is realized by the framework proposed in this paper.

**Figure 2 sensors-22-04697-f002:**
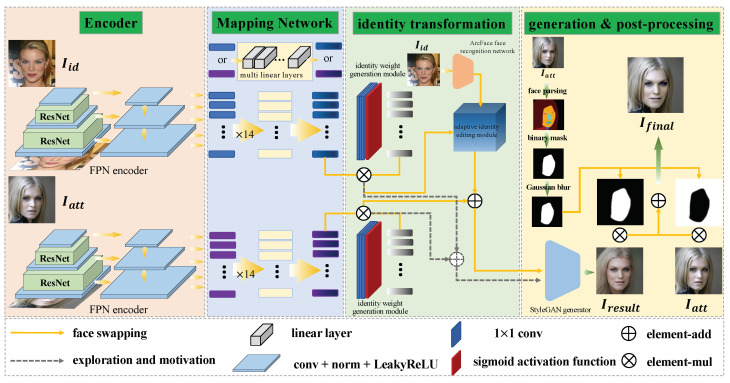
The overall structure of the network, which is composed of feature pyramid network with ResNet, mapping network, identity weight generation module, adaptive identity editing module, generator, and post-processing process.

**Figure 3 sensors-22-04697-f003:**
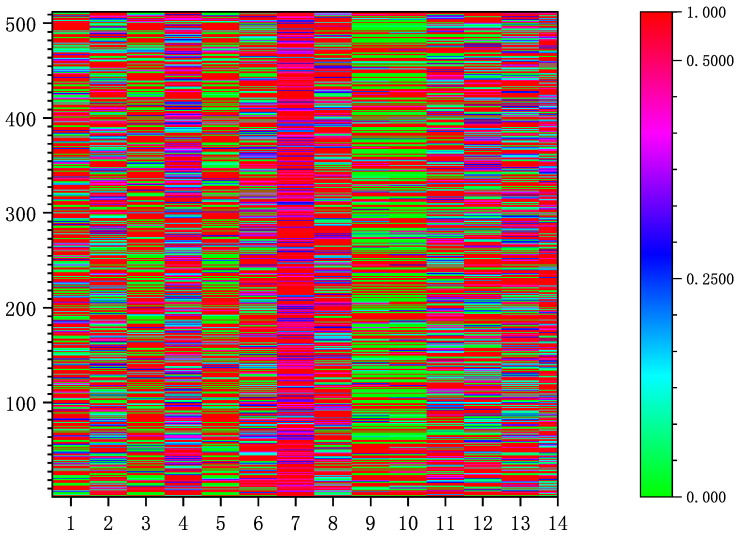
The visualization result of the output value of the identity weight generation module.

**Figure 4 sensors-22-04697-f004:**
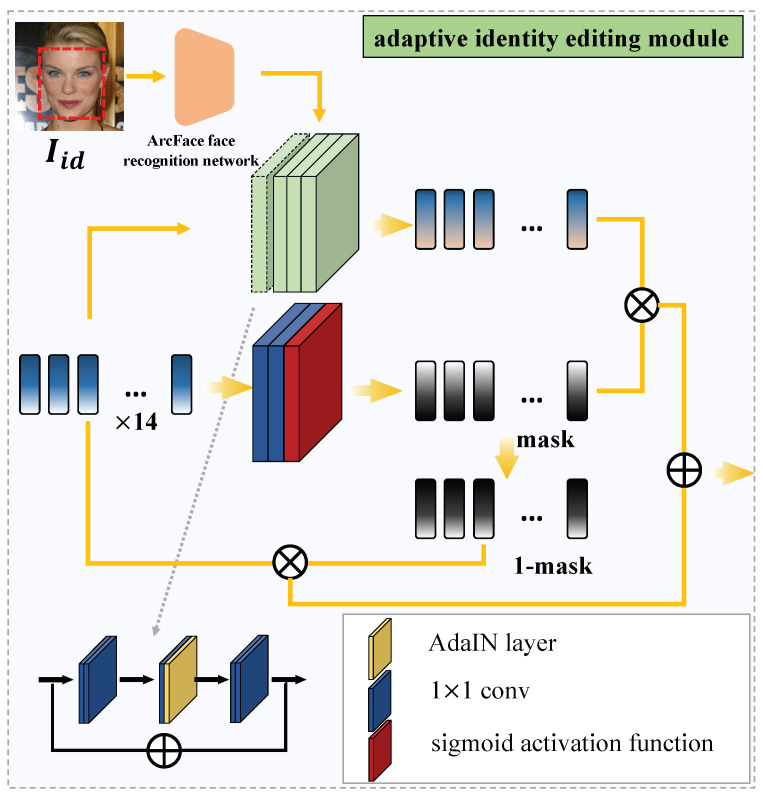
The structure of the adaptive identity editing module.

**Figure 5 sensors-22-04697-f005:**
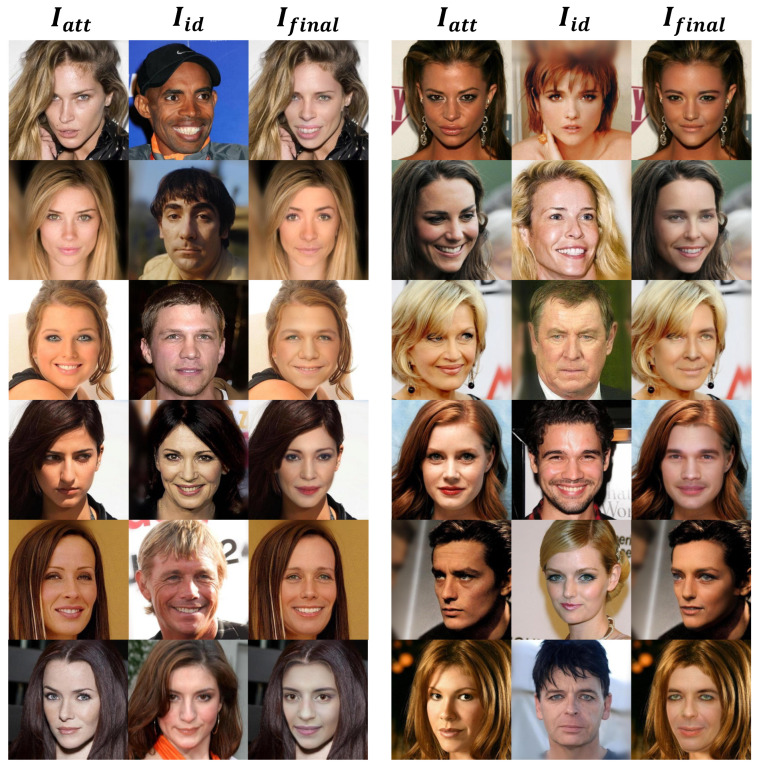
Partial results of our framework.

**Figure 6 sensors-22-04697-f006:**
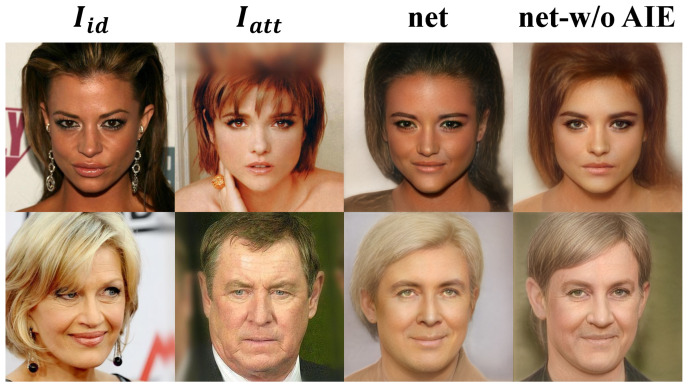
The results of the experiment to verify the superiority of the adaptive editing module.

**Figure 7 sensors-22-04697-f007:**
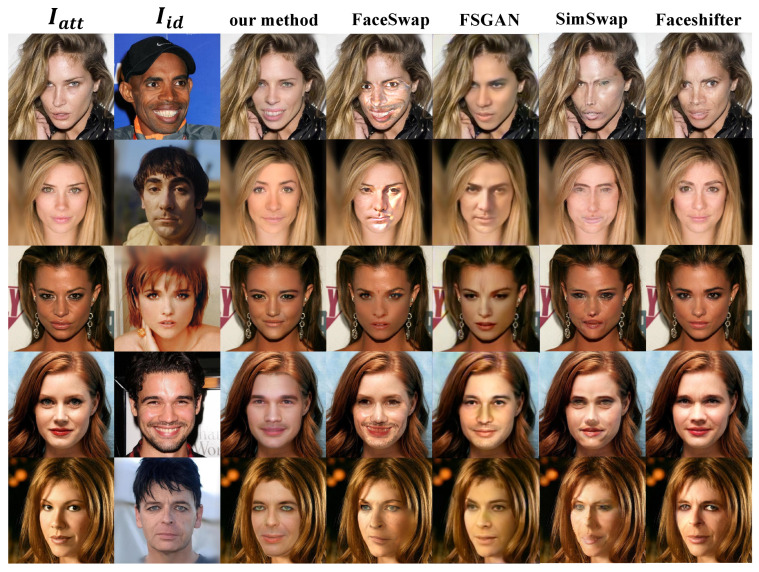
Compared with other methods. The comparison methods consist of FaceSwap, FSGAN, SimSwap, and Faceshifter.

**Table 1 sensors-22-04697-t001:** Quantitative analysis of the superiority of the adaptive identity editing module. Bold represents the optimal value.

Method	Id Similarity ↑	Exp Similarity ↓	FID ↓
net−w/oAIE	0.57	0.25	**58.7864**
net	**0.58**	**0.23**	58.8624

**Table 2 sensors-22-04697-t002:** Quantitative analysis of the comparative experiment with other method. Bold represents the optimal value.

Method	Id Similarity ↑	Exp Similarity ↓	FID ↓
FaceSwap	0.37	3.32	216.78
FSGAN	0.45	1.64	67.54
SimSwap	0.54	1.31	69.84
Faceshifter	0.51	**0.19**	58.9625
Ourmethod	**0.58**	0.19	**58.8624**

## Data Availability

Not applicable.

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
