# Peer review of "Enriching Facial Anti-Spoofing Datasets via an Effective Face Swapping Framework"

_sensors, 2022, doi:10.3390/s22134697_

Round 1

Reviewer 1 Report

In this work, the authors propose an efficient face swapping framework which can generate photo-realistic results. An identity weight generation module, an adaptive identity editing module and a post-processing method are utilized in the proposed framework authenticity of the generated results. The proposed framework is evaluated on CelebA-HQ dataset. And according to the experiment results in Table 2, the proposed method can achieve comparative or better performance compared with other methods.

For my understanding, the proposed method performs well because of the identity weight generation module, the adaptive identity editing module and the post-processing method are utilized. I suggest the authors to include more discussions on the novelty of the identity weight generation module, the adaptive identity editing module and the post-processing method and explain why the proposed method can perform better than other methods.

Author Response

Dear Editor and Reviewers,

Thanks very much for taking your time to review this manuscript. We really appreciate all your comments and suggestions! We have carefully considered the suggestion of Reviewer and make some changes.

The following is our reply to the comments:

Reviewer1:

In this work, the authors propose an efficient face swapping framework which can generate photo-realistic results. An identity weight generation module, an adaptive identity editing module and a post-processing method are utilized in the proposed framework authenticity of the generated results. The proposed framework is evaluated on CelebA-HQ dataset. And according to the experiment results in Table 2, the proposed method can achieve comparative or better performance compared with other methods.

For my understanding, the proposed method performs well because of the identity weight generation module, the adaptive identity editing module and the post-processing method are utilized. I suggest the authors to include more discussions on the novelty of the identity weight generation module, the adaptive identity editing module and the post-processing method and explain why the proposed method can perform better than other methods.

Answer:

We are so grateful for your kind suggestion. We have made the following changes to this suggestion. (1) For ‘identity weight generation module’, we make the following detailed explanation.

It is well known that the latent codes of an image contain all the features of the corresponding image. These features can be roughly divided into two categories: fine-grained features, such as identity information, expression information, posture information, and coarse-grained features, such as color information, structure information, etc. In the latent space, the specific location of these features is uncertain, and the features are entangled with each other, which causes a great degree of confusion. In order to edit only the features that we want to edit and prevent the interference of other features as much as possible, we need to extract the area to be edited. At this stage, researchers have done too little research on the structural level of latent space, so they can not accurately determine the specific location of the corresponding features. Therefore, we propose an identity weight generation module, which can adaptively find the location of the area to be edited through a large number of data training and weight updating.

(2) For ‘adaptive identity editing module’, we make the following detailed explanation.

In order to realize the transformation of image information, we propose the adaptive identity editing module. The module uses the identity information extracted from the existing face recognition network to change the specific information in the latent codes. We use the identity information extracted from the face recognition network to fix the direction of the latent codes. The adaptive mask in this module further extracts the feature regions that need to be edited and reduces the overall effect decline caused by changes to other latent features.

(3) For ‘post-processing’, we make the following detailed explanation.

In order to achieve high-resolution face generation, we use StyleGAN's generator as the face generator of the method proposed in this paper. However, there is a defect in using the prior knowledge of the face, that is, the generated images only focus on the face region, and often ignore the hair region and the background region. This greatly reduces the authenticity of the generated image. To solve the above problems, we use the existing face parsing to get the face semantic segmentation mask, use the mask to weight the generated image and the image that provide other attributes, and finally get the photo-realistic results.

Through the identity weight generation module and the adaptive identity editing module, we have realized the accurate positioning of the area to be edited, assisted by the most advanced face recognition network, guided the editing direction of potential features, and realized the accurate change of identity information. Finally, through post-processing, the hair area and background area are accurate. Integrating each module proposed in this paper, the overall effect is finally achieved, and the advantages are achieved in comparison with other methods.

We added the above explanations to the revised version and marked it in yellow in the following documents. The reviewer's suggestions are of great help to improve the quality of our paper. Thanks again!

Reviewer 2 Report

The paper details a novel method for swapping the identity of facial images, with the purpose of enriching datasets for anti-spoofing detection. The authors present a novel weighting method for the part of the latent space that encodes identity, as well as a ne method for editing the identity using adaptive instance norm. The authors also present their own post-processing formula to improve the fidelity of the generated images.

The manuscript's language and style are acceptable, although the english should be improved. There are numerous cases of incorrect use of singular/plural cases or definite/indefinite articles (a/an/the). The manuscript also contains a few oddly short sentences that should be rephrased to improve readability. Examples: "Enriching facial spoofing dataset is an effective method." To do what, exactly?

The literature review is contains most of the relevant work, but the style and presentation resembles a bullet point list simply mentioning these advances. A proper review should elaborate the fundamentals of the previous work at least a little bit to make later parts of the paper more understandable, while also serving to distinguish the authors' own contributions.

The manuscript's presentation of the novel method is somewhat difficult to read. The authors sometimes reference things they detail in later chapters (such as the loss functions in section 3.1) or haven't introduced properly. I had the following questions regarding how the proposed model works. I had to re-read section 3.3 a few times to (mostly) understand what the authors did and why. Also, the description of the self-reconstruction loss in section 3.5.2 is nonsensical.

In the final part of the manuscript the authors present a quite well-executed ablation study between their method and other SOTA works. This comparison is well-written and conclusive, supporting the authors' claim to have created a superior method. Lastly, I have the following concrete suggestions and questions:

1. Throughout the paper, the authors have an odd fixation with facial portraits in healthcare systems. Wouldn't openly accessible images (such as the ones on social media) be more vulnerable to abuse?

2. A few times the authors claim to have created an "efficient" method. I am reasonably sure they meant "effective". The paper doesn't present any experiments regarding efficiency, so such claims should not be made.

3. To introduce the contents of a section the authors should write "In this section, we will present..."

4. What are "no-learned" methods (line 104)? Non-learning methods maybe?

5. Are there two identity weight generator networks? Or how does it know which image is the identity image and which is the attribute one?

6. If the "identity distance" increases as the identities become more similar, shouldn't it be called "identity similarity"?

Author Response

Dear Editor and Reviewers,

Thanks very much for taking your time to review this manuscript. We really appreciate all your comments and suggestions! We have carefully considered the suggestion of Reviewer and make some changes.

The following is our reply to the comments:

Reviewer2:

The paper details a novel method for swapping the identity of facial images, with the purpose of enriching datasets for anti-spoofing detection. The authors present a novel weighting method for the part of the latent space that encodes identity, as well as a new method for editing the identity using adaptive instance norm. The authors also present their own post-processing formula to improve the fidelity of the generated images.

The manuscript's language and style are acceptable, although the english should be improved. There are numerous cases of incorrect use of singular/plural cases or definite/indefinite articles (a/an/the). The manuscript also contains a few oddly short sentences that should be rephrased to improve readability. Examples: "Enriching facial spoofing dataset is an effective method." To do what, exactly?

Answer:

Thank the reviewer for pointing out the problem. To solve this problem, we have made the following modifications.

(1) We revised “Taking the users’ portrait of multimedia system as an example” to “Taking the users’ portraits of multimedia system as examples”, revised “personal portrait is facing malicious tampering and forgery, which poses a potential threat to personal privacy security and social impact” to “personal portraits are facing malicious tampering and forgery, which pose a potential threat to personal privacy security and social impact”, revised “Enriching Facial anti-spoofing dataset is an effective method” to “Enriching Facial anti-spoofing dataset is an method to solve above problem”, revised “which has brought unprecedented changes to many fields.” to “which have brought unprecedented changes to many fields”, revised “Many countries and regions actively carry out the improvement of relevant regulations and laws.” to “Many countries and regions actively have carried out the improvement of relevant regulations and laws.” in page1. We revised “which is based on AdaIN mechanism and can realize identity transformation” to “which is based on the AdaIN mechanism and can realize the identity transformation”, revised “We use identity weight generation module to increase the attention to the identity information. to “We use the identity weight generation module to increase the attention to the identity information.” in page 2. We revised “which poses a potential threat to personal privacy and public opinion” to “which pose a potential threat to personal privacy and public opinion” and revised “researchers found that editing images through latent codes can achieve better results” to “researchers found that editing images through latent codes could achieve better results” in page 3. We revised “The weight generation module is composed of multiple 1 × 1 convolution and sigmoid activation function” to “The weight generation module is composed of multiple 1 × 1 convolution and the sigmoid activation function” and revised “The structure is based on AdaIN mechanism and adaptive blending mechanism.” to “The structure is based on the AdaIN mechanism and the adaptive blending mechanism.” in page 6. We revised “And its process is as follows:” to “And the process is as follows:” in page 7. We revised “we can see that although the generation result” to “we can see that although the generation results” in page 10. We revised “the image generated by net-w/o AIE is not as good as net in identity transformation” to “the images generated by net-w/o AIE are not as good as net in identity transformation” in page 11.

The literature review is contains most of the relevant work, but the style and presentation resembles a bullet point list simply mentioning these advances. A proper review should elaborate the fundamentals of the previous work at least a little bit to make later parts of the paper more understandable, while also serving to distinguish the authors' own contributions.

Answer:

Thank the reviewer for pointing out the problem. We have added fundamentals of the previous work to the Related Work, that is, the introduction of Generative Adversarial Network, which will play an important role in understanding the work of this paper.

The manuscript's presentation of the novel method is somewhat difficult to read. The authors sometimes reference things they detail in later chapters (such as the loss functions in section 3.1) or haven't introduced properly. I had the following questions regarding how the proposed model works. I had to re-read section 3.3 a few times to (mostly) understand what the authors did and why. Also, the description of the self-reconstruction loss in section 3.5.2 is nonsensical.

Answer:

Thank the reviewer for correcting the problems in the Method part. We reviewed our manuscript and answered the above questions as follows

(1) We added an index in 3.1, and corrected the questions raised by the reviewer in 3.4.

(2) For the problem in subsection 3.3, we believe that it is caused by the deviation of the identity weight generation module in subsection 3.1 and subsection 3.3. Thank you very much for your comments on this subsection. We explained the module in detail in 3.3 and elaborated the differences from 3.1. Then, we reorganized the structure of subsection3.3 to make it clearer.

(3) For the self-reconstruction loss, we are sorry we didn't explain it clearly. We re-explain the self-reconstruction loss as follows:

Because there is no the ground truth in the face swapping task, there is no strong constraint in the training process. In order to enhance the robustness and stability of the network, we propose the self-reconstruction loss. The process of self-reconstruction loss: set  and  to the same image, and the final result of network output should be the same as that of input. Through the above process, we can achieve strong constraints in the training network, which greatly improves the stability and robustness of network training. Could you agree with our reinterpretation?

In the final part of the manuscript the authors present a quite well-executed ablation study between their method and other SOTA works. This comparison is well-written and conclusive, supporting the authors' claim to have created a superior method. Lastly, I have the following concrete suggestions and questions:

  1. Throughout the paper, the authors have an odd fixation with facial portraits in healthcare systems. Wouldn't openly accessible images (such as the ones on social media) be more vulnerable to abuse?

Answer:

Thank the reviewer for raising this question. This is due to our consistent focus on the privacy of healthcare system. In the revised version, we will expand the scope of application to other areas such as social media.

  1. A few times the authors claim to have created an "efficient" method. I am reasonably sure they meant "effective". The paper doesn't present any experiments regarding efficiency, so such claims should not be made.

Answer:

Thank the reviewer for their comments on this issue. We highly appreciate the reviewers' suggestions. In the revised version, we no longer claim “efficient”.

  1. To introduce the contents of a section the authors should write “In this section, we will present...”

Answer:

Thank the reviewer for pointing out this problem. We checked the full text and corrected the errors.

  1. What are "no-learned" methods (line 104)? Non-learning methods maybe?

Answer:

Thank for the reviewer’s review of the details of the paper. We agree with the reviewer. We think “non-learning” is more appropriate. In the revised version, we have made corresponding modifications.

  1. Are there two identity weight generator networks? Or how does it know which image is the identity image and which is the attribute one?

Answer:

Thanks for your questions. In the network, there are two identity weight generation modules. They have the same structure. They are responsible for extracting the regions of interest of identity information and attribute information respectively. We are sorry we didn't explain it clearly in the original text. We make corrections in the revised version.

  1. If the “identity distance” increases as the identities become more similar, shouldn't it be called “identity similarity”?

Answer:

Thank the reviewer for their comments on this detail. We quite agree with this view. We change the “identity difference” to “identity similarity”. At the same time, we change “expression difference” to “expression similarity”. We believe that the function of the quantitative metrics are more clearly represented after the change.

We added the above explanations to the revised version and marked it in green in the following documents. The reviewer's suggestions are of great help to improve the quality of our paper. Thanks again!